# Instability of Traveling Pulses in Nonlinear Diffusion-Type Problems and Method to Obtain Bottom-Part Spectrum of Schrödinger Equation with Complicated Potential

**Michael I. Tribelsky** [1,2]

1 Faculty of Physics, M. V. Lomonosov Moscow State University, 119991 Moscow, Russia; mitribel@gmail.com
2 National Research Nuclear University MEPhI, Moscow Engineering Physics Institute, 115409 Moscow, Russia

**Abstract:** The instability of traveling pulses in nonlinear diffusion problems is inspected on the example of Gunn domains in semiconductors. Mathematically, the problem is reduced to the calculation of the "energy" of the ground state in the Schrödinger equation with a complicated potential. A general method to obtain the bottom-part spectrum of such equations based on the approximation of the potential by square wells is proposed and applied. Possible generalization of the approach to other types of nonlinear diffusion equations is discussed.

**Keywords:** nonlinear diffusion; traveling waves; stability; Goldstone modes; Schrödinger equation; spectrum of low-exited states





## 1. Historical Remarks

When the Editors kindly offered me to submit a paper to this Special Issue dedicated to my fifty years in physics, I began to think about a possible topic of the paper. Finally, I decided that the best is to generalize the results of my very first paper [1], which formally was published exactly fifty years ago. I said "formally" because *actually* this paper has never been published. Perhaps, its story is so remarkable that it is worth telling it here.

The point is that though I graduated from the Lomonosov Moscow State University (MSU)—the one where now I head a laboratory—I did not enter this university in the usual, standard manner. It so happened that the university I entered was the Belorussian State University (BSU) in Minsk. Now, Minsk is the capital of independent state Belarus, while, at that time, Minsk and Moscow both belonged to a single state: the Soviet Union. In Minsk, I met my first scientific adviser Mikhail Aleksandrovich El'yashevich [2].

Then, upon completing my first two university years in Minsk, I moved to Moscow. Thus, I became a student of MSU due to my transfer from BSU. Just one letter difference in the names meant the drastic difference in the ranks. Though BSU was quite a good university, MSU was (and is) the Number One.

Doing paperwork related to the transfer, I asked El'yashevich for a reference letter to one of his collaborators in Moscow. I then obtained a letter to his former Ph.D. student Sergei Ivanovich Anisimov [3], who became my next scientific adviser.

It was 1969. At that time, I could not even imagine how lucky I was. Anisimov was employed by the Landau Institute for Theoretical Physics. The Institute was created just five years ago to collect "under a single roof" the first generation of Lev Davidovich Landau's disciples [4]. By the time I am talking about, all of them had become first magnitude stars in the scientific sky.

Thus, suddenly and almost by chance, I became embedded in the scientific atmosphere representing the very top of theoretical physics in the USSR, and I would say in the entire world too. Moreover, I had even more good luck, though, I did not know it yet: In the very same year of my transfer to MSU, a prominent theoretical physicist Il'ya Mikhailovich Lifshits [5] succeeded the late Landau's position of the Head of the Theoretical Physics

Department at the Kapitza Institute [6]. To this end, he moved to Moscow from Khar'kov (a big Ukrainian city), where he resided before. In addition to this position, Il'ya Mikhailovich got a professorship at the Chair for Quantum Theory, the Faculty of Physics, MSU. By that time, another employee of the Landau Institute and a disciple of Il'ya Mikhailovich, namely, Mark Yakovlevich Azbel [7], already shared his position at the Landau Institute with a professorial position of this Chair. The second disciple of Il'ya Mikhailovich, who came from Khar'kov to Moscow and became a Professor of the same Chair, was Moisei Isaakovich Kaganov. Among other scientific accomplishments of this group was the galvanomagnetic theory of electrons with an arbitrary dispersion law. The theory describes the effects of both electric and magnetic fields acting together on free electrons in metals. In this theory, electrons are regarded as quasi-classical particles, but, instead of the conventional dependence of the energy on (quasi)momentum $\varepsilon(p) = p^2/(2m)$, this dependence may be arbitrary. The theory was a breakthrough in quantum solid-state physics, and was named after its creators—the LAK theory (Lifshitz, Azbel, Kaganov).

There is an interesting story related to this abbreviation. When another one of Landau's disciples, Alexander Solomonovich Kompaneetz, known, in addition to his outstanding scientific results, for his sense of humor, leant about LAK theory, he said, "It is excellent that the authors did not employ the inverted order of them." The joke is that *kal* in Russian means excrements.

To complete my description of the Chair for Quantum Theory, I should add that it was headed by one of the most prominent experts in theoretical physics, a very respectable person with the highest moral standards, Academician of the Soviet Academy of Sciences, Mikhail Aleksandrovich Leontovich [8]. Alas, all of them have already passed away.

In 1969, I knew nothing about these people and the Chair, but Anisimov did know. Therefore, when I asked his advice about the specific Chair at the Faculty of Physics for my specialization, he immediately replied, "The one where I.M. Lifshitz is a Professor." I took his advice and applied for the specialization at this Chair. Once again, I was lucky — my application was approved, and in addition to the excellent external scientific environment at the Landau Institute, I benefited from that at the Chair for Quantum Theory.

Soon after my appearance at the Chair, I began to attend lectures on the quantum theory of metals given by Kaganov. Bearing in mind that Kaganov was a brilliant lecturer, it is easy to understand that I admired the beauty of the lectures and that of the theory as a whole. Thus, it is easy to understand that, when Anisimov asked me about the preferences for the topic of the future study, my reply was, "Something from quantum solid-state physics".

It is worth mentioning that, at the time, I did not have any idea about the specific subfield, where the accomplishments of Anisimov lay (namely, laser–matter interaction, physical hydrodynamics, shock waves, plasma physics, and the like). Fortunately, he was a physicist with broad interests and understood physics far beyond the frames of his own subfield. It was a typical feature of physicists from the Landau Institute originated by Landau himself: Broad knowledge helps to see cross-links between different, seemingly unrelated, problems. This, in turn, sometimes helps to obtain very beautiful and unexpected results.

Then, according to my desire, Anisimov posed me a problem from quantum solid-state physics. It was related to the Gunn effect in semiconductors [9]. At that time, the effect was a fascinating, challenging topic, and, up to now, it still attracts a great deal of attention from researchers [10–15].

Naturally, now the understanding of the effect is more profound, and its mathematical description is much more elaborated than it was 50 years ago; see, e.g., Ref. [16]. However, since the goal of this paper is to generalize the methods and results discussed in Ref. [1], making them applicable to a broad class of related problems, rather than to inspect specific peculiarities of the Gunn effect itself; in what follows, I stick to the old model of the effect [17] employed by Knight and Peterson [18] and then, in my paper [1].

Very briefly, the essence of the phenomenon is as follows. In a strong enough electric field, **E**, applied to a semiconductor, the conductivity of the sample depends on **E**, and the current–voltage curve becomes nonlinear. In some cases, calculations based on the assumption of the spatially homogeneous distribution of the current density, **j**, and **E** along and across the sample give rise to very unusual behavior of this dependence so that, in a certain area of the **E** values, an increase in **E** results in a *decrease* in **j**.

In what follows, only one-dimensional cases will be considered, so I can replace $\mathbf{j}(\mathbf{E}) \rightarrow j(E)$. Then, by definition, the conductivity $\sigma = j/E$. Let us define the differential conductivity as $\sigma_d = \mathrm{d}j/\mathrm{d}E$. Thus, the area mentioned above is characterized with a negative differential conductivity. Here, I will not discuss the microscopic mechanisms explaining the negativeness of $\sigma_d$; a detailed description may be found, e.g., in Ref. [19]. Further increase in $E$ makes $\sigma_d$ positive again so that the overall shape of the current–voltage curve resembles letter "*N*", see Figure 1.

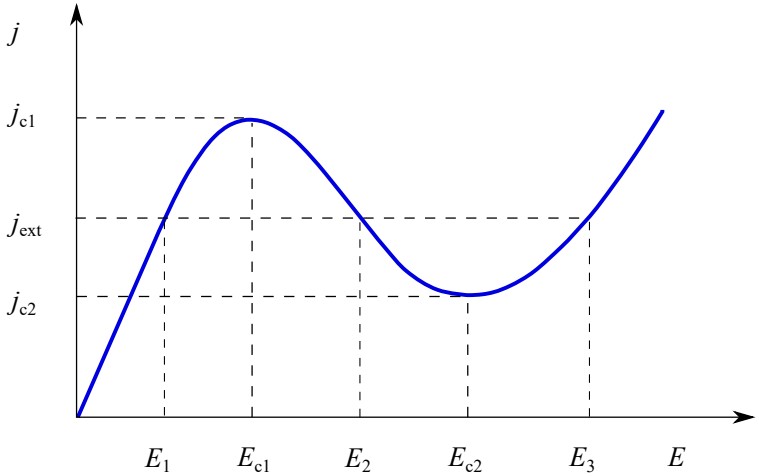

**Figure 1.** A letter-*N*-shape current–voltage characteristics obtained under the assumption that electric field $E = const$ along and across the sample: the differential conductivity, $\sigma_d$ is negative at $E_{c1} < E < E_{c2}$.

It occurs that the assumption about the spatially uniform distribution of $j$ and $E$ in the regions with $\sigma_d < 0$ is erroneous. This distribution is unstable against small spatially-inhomogeneous perturbations and, eventually, is destroyed owing to their growth. In certain cases, the instability ends up forming a strong field domain bounded by the corresponding layers of charge density. The domain drifts along the sample with a constant speed until it hits the sample edge (anode). The domain disintegrates there, a new one emerges at the opposite side of the sample, and the process repeats. As a result, oscillations with the period $L/v$ are generated. Here, $L$ stands for the sample length and $v$ is the drift speed of the domain. This is the Gunn effect [9]. It is successfully used in Gunn diodes to generate microwave oscillations [19].

Let us consider an idealized case of a single traveling strong-field Gunn domain drifting with a constant velocity along an infinitely-large sample. The "strong-field domain" means that the field outside it equals $E_1$ (see Figure 1), while inside the domain, it is greater than that. Then, if the voltage applied to the sample is constant, the single domain with a fixed shape is stable, while a configuration with several domains is not. However, if an external source fixes the current in the sample, even the single domain becomes unstable. The instability affects the faces of the domain, which begin to move in opposite directions with respect to the center of the traveling domain. If they move to each other, the domain contracts and, eventually, collapses. If the faces move in the opposite direction, the domain expands and transforms into two traveling layers [19].

Linear analysis of this secondary instability of a single traveling domain at a fixed current in the circuit was performed by Knight and Peterson [18]. Mathematically, the

stability problem was reduced to the calculation of a gap between the ground and the first excited states in the one-dimensional Schrödinger equation with a complicated potential (see below). To this end, Knight and Peterson employed the Wentzel-Kramers-Brillouin (WKB) approximation. However, this approximation is accurate for highly excited states when the characteristic spatial scale of the wave function oscillations is small relative to the one for the variations of the potential. This is not the case for the ground state. Hence, the accuracy of the results obtained in Ref. [18] through the WKB method, at least, was questionable. The problem, posed for me by Anisimov, was to check the results of Knight and Peterson employing for the calculations an approximation different from WKB.

If I faced this problem now, quite probably, I would have used the Ritz method supplemented by the orthogonality condition of the wave functions of the ground and excited states [20]. However, at that time, I was much more ignorant than I am now. Therefore, instead of taking a simple, known way (perhaps, at that time, it was neither simple nor known for me), I decided to go on my own one. Specifically, I decided no less than to find a new method to obtain approximate solutions to the Schrödinger equation opposite to the WKB-method, which could be suitable for the ground and low-excited states. Furthermore, I succeeded in doing that! So, maybe, ignorance is not always bad.

The main idea of the developed approach is somewhat unusual for quantum mechanics, where approximations conventionally are targeted to a wave function, while the potential is given and fixed. However, if one has ground and low-excited states in a complicated potential, the potential has a sharply varying profile relative to that of the wave functions, and the latter is not very sensitive to the fine details of the former. If so, why does one not try to approximate *the potential*, with some simple shapes, say, with square wells? Then, the Schrödinger equation becomes either exactly solvable or readily treated by perturbation methods.

The most challenging task was to set the first step in this way. The rest was just a matter of not so complicated calculations. I quickly did them and presented the results to Anisimov. "Very well," he said, "the problem is solved. Write a paper. One more point to be made. Il'ya Mikhailovich Lifshitz has organized a periodic scientific seminar at your Faculty. It takes place every second and fourth Thursday of a month from September to June. I recommend you to contact Il'ya Mikhailovich and ask him to put your talk about this study in the seminar program.".

Up to now, I remember how difficult it was for me (a fourth-year undergraduate student) to approach such a famous scientist as Il'ya Mikhailovich was and request a talk at his seminar. Finally, I gathered up all my courage and did it. "Excellent," replied Il'ya Mikhailovich, "Please contact the seminar's secretary, Mr. Rzhevsky, and ask him to find the nearest free spot in the program. Will 45 minutes be enough for you?".

To give a 45-min talk in front of an audience of top-rank experts, including a dozen of world-class scientists! My knees turned to jelly, but there was no way to retreat.

It is remarkable that I can vividly remember any moment before and after my talk, but nothing of the talk itself. However, it seems that I stood this test. Moreover, the talk at this seminar was a milestone for my relationship with Il'ya Mikhailovich. Since then, every one of my new results was discussed with him, either through a talk at the seminar or in a private manner at his office at the Kapitza Institute. Later on, I became a close associate and coauthor of Il'ya Mikhailovich [21]. We even had a joint Ph.D. student. Our close contact lasted until his unexpected premature death of a heart attack in 1982.

However, all this will be later on. At the time I am talking about, I could not imagine even a small part of that.

Thus, the first task (the talk) was complete, but the second remained: I had to write a paper. It was my very first paper, and it took a lot of my time and efforts to do that. Finally, an extended manuscript (in Russian) was submitted to *Fizika i Tekhnika Poluprovodnikov (Physics and Technology of Semiconductors)*.

It was the beginning of the bad luck for this paper. At that time, a new publication option was introduced. For some papers (especially lengthy ones), only abstracts were published. The papers themselves were *deposited* in specially assigned institutions. If an abstract of such a paper drew somebody's attention, and he/she was interested in the complete text, a copy of the one could be posted to him/her upon a request. Maybe it was an attempt to reduce the printed size of scientific journals and to solve, at least partly, the eternal Soviet problem of paper deficit. Anyway, my paper was accepted under this condition. Its abstract was published, see Figure 2, while the full text was deposited at *Research and Development (R&D) Institute Electronics*.

Since then, many events have occurred. The country named the Soviet Union does not exist anymore. Regarding *R&D Electronics*, I am afraid it has shared the destiny of the country. Now, I am residing within walking distance from the building where *R&D Electronics* used to be. It is a shopping mall there. Then, it is quite probable that the full text of my paper has ended up in a nearby scrap-heap, and the abstract reproduced in Figure 2 is the only remaining piece of the paper.

> Вып. 10           ДЭ—416 от 30 июня 1971 г.
>
> ## ОБ ИНКРЕМЕНТЕ НЕУСТОЙЧИВОСТИ ГАННОВСКИХ ДОМЕНОВ В РЕЖИМЕ ПОСТОЯННОГО ТОКА
>
> ### М. И. Трибельский
>
> Исследуется развитие неустойчивости в широких ганновских доменах (ширина вершины много больше ширины фронтов) в режиме стационарного внешнего тока. В основу расчетов положена феноменологическая модель, в которой полный ток складывается из тока проводимости и диффузионного тока. Неустойчивость заключается в том, что фронты домена начинают смещаться в противоположных направлениях. Вычислен инкремент нарастания неустойчивости. Коэффициент диффузии предполагался независящим от поля. Задача математически свелась к решению уравнения типа одномерного уравнения Шредингера с некоторым сложным потенциалом. Показано, что результаты слабо зависят от вида этого потенциала. Поэтому он был заменен потенциалом в виде двух прямоугольных ям, что позволило выполнить расчеты в конечном виде. Задача решена в двух предельных случаях: симметричного и сильно несимметричного доменов. В обоих случаях вычислены дополнительное падение напряжения на домене, а также ширина фронтов и вершины домена.
>
> Существенно, что в конечный результат не входят какие-либо интегральные характеристики. Он зависит только от значения функций $j(E)$ и $dj/dE \equiv \sigma_d$ в некоторых характерных точках. Развитый метод без существенных изменений может быть применен к исследованию характера неустойчивости волн в виде двух и более доменов.
>
> Поступило в Редакцию
> 13 марта 1971 г.

**Figure 2.** The only ever published piece of paper [1] (in Russian).

The English translation reads:

<div align="center">

Vol. 10 DE-416 dated 30 June 1971
On the increment of the instability of the Gunn domains in the direct current regime
M. I. Tribel'skii

</div>

The growth of instability of wide Gunn domains (the width of the top is much larger than the widths of the faces) at the stationary external current regime is inspected. The basis of calculations is the phenomenological model, in which the total current is composed of the conductivity current and the diffusion one. Instability affects the domain faces so that they begin to shift in opposite directions. The instability increment is calculated. The diffusion coefficient is supposed to be independent of the field. Mathematically the stability problem is reduced to a one-dimensional Schrödinger equation with a certain complicated potential. It is shown that the results are weakly dependent on details of this potential. Therefore, the potential is approximated by two square potential wells (separated

by a barrier), which made it possible to obtain an explicit expression for the increment. The problem is solved in two limiting cases, namely symmetric and highly asymmetric domains. In both cases, additional drops of voltage on the domain are calculated, as well as the width of the faces and the top of the domain.

It is essential that the final result does not include any integral characteristics of the problem. It depends only on the value of the functions $j(E)$ and $dj/dE \equiv \sigma_d$ at certain characteristic points. A developed method without significant changes may be extended to the study of the instability of waves in the form of two or more domains.

Received 13 March 1971.

The next attempt to publish these results I made after the defense of my Master Science Thesis. An appointed referee of the thesis was another disciple of Landau, Igor Ekhiel'evich Dzyaloshinskii [22]. After reading the thesis, he said that the results of this level should be available to the international community, and I should publish them abroad in English (when the first draft of the present paper was ready, I learned the sad news: Igor Ekhiel'evich Dzyaloshinskii passed away on 14 July 2021).

To publish abroad, ... it was easier to say than to do. Not to mention poor English, which I had at that time, sending a scientific paper abroad from the Soviet Union was not simple at all. The authors themselves were not eligible to do that. A manuscript had to be sent through specially authorized personnel. The personnel decided whether or not the paper could be submitted abroad, and, if the decision was affirmative, they took care of the submission.

Moreover, prior to the acceptance of the manuscript by the personnel, the authors had to do plenty of paperwork. On top of that, it took 2–3 months on average for mail to be delivered to the addressee. Up to now, I wonder why this was so much. Even if horses delivered the mail; it would not have taken such a long time!

I discussed the matter with Anisimov, and we decided to submit the paper to the East-German journal *Physica Status Solidi* published in English. There were two reasons for this choice. First, sending a paper to an Eastern bloc country required less paperwork, and chances to get permission for the submission were higher than that in the case of a Western journal. Second, the requirement for the English quality in this journal was not as strict as those in the West. The latter was important since my English was far from being perfect.

Thus, I wrote in English an elaborated version of Ref. [1] including some new results, did all the required paperwork, gave the bunch of documents to the "authorized personnel," and... lost control over the submission. Half a year elapsed, but I had not heard anything from the Editors. Then, I sent a postcard to *Physica Status Solidi* asking for the status of my paper. A reply came surprisingly fast—in just four months. However, it was pretty unexpected. The Editors informed me that they had never received my manuscript.

By that time, on the one hand, I had already published a paper [23], where the secondary instability of the Gunn domain was inspected just employing the Ritz method. On the other hand, I got a job and, owing to that, was forced to abandon my study in solid-state physics and focus on an entirely different topic.

Eventually, the results discussed in Ref. [1] have remained unpublished. Now, fifty years later, I try to realize the advice of Dzyaloshinskii and make these results available to the international community. Perhaps fifty years is a too long period to complete a task, but "that is not lost that comes at last!".

At the end of these, perhaps lengthy, remarks, I have to say that the results discussed below are not exactly the same as those in Ref. [1]. First, it is not good to publish the same results twice, even if the fifty years lie between the two publications. Second, I could not do this, even if I wanted to—the original manuscript is lost, and I do not remember all details. Last but not least: now I am a bit more experienced and educated than I was fifty years ago. Therefore, I extracted from this old problem the essential points and generalized them. These points are as follows: (i) the conclusion about the instability of traveling pulses in a broad class of nonlinear diffusion-type problems and (ii) a new method to obtain

the bottom-part spectrum of the Schrödinger equation with a complicated potential. A discussion of these two issues is given below.

## 2. Problem Formulation

Thus, the problem is to find the instability increment for a single traveling Gunn domain at a fixed current in the circuit. According to what has been said above, the current–voltage characteristic of the semiconductor sample in question has the shape schematically shown in Figure 1. Regarding the external current, $j_{ext}$, let us suppose that it satisfies the restrictions $j_{c1} < j_{ext} < j_{c2}$, so that the equation $j(E) = j_{ext}$ always has three roots, $E_{1,2,3}$.

It is important to stress that the curve, shown in Figure 1, is not the *actual* current–voltage characteristic of the sample. As mentioned above, it would have been the one provided $E$ is a constant along and across the sample. Obviously, this is not the case for the traveling domain, when $E$ is coordinate- and time-dependent. Therefore, only the stable branches of the presented curve with $\sigma_d > 0$ coincide with the actual current–voltage characteristic. In contrast, the whole curve in Figure 1 should be regarded as the field dependence of the normalized average electron drift velocity [17].

It is convenient to normalize the electric field over $E_2$ and $j(E)$ over $j_{ext}$ introducing the dimensionless quantities $\mathcal{E} \equiv E/E_2$ and $u(\mathcal{E}) \equiv j(E)/j_{ext}$. Then, under certain assumptions, in the traveling coordinate frame connected with the domain, the normalized electric field in the sample is described by the following equation [19]:

$$\mathcal{D}\mathcal{E}_{\xi\xi} + \alpha[s - u(\mathcal{E})]\mathcal{E}_\xi + [1 - u(\mathcal{E})] = \mathcal{E}_\eta(\xi, \eta), \tag{1}$$

where the subscripts indicate the corresponding derivatives. Equation (1) is written in dimensionless variables, whose detailed definition is not important for the subsequent analysis (it may be found in Ref. [19]). Note only that $\mathcal{D}$, $s$, $\xi$, and $\eta$ stand for the diffusion coefficient, the domain velocity in the laboratory coordinate frame, traveling coordinate, and time, respectively; $\alpha = const > 0$ is the ratio of two characteristic spatial scales of the problem at $E = E_2$.

Let us suppose that $\mathcal{D} = const$. This assumption simplifies calculations, but it is not crucial for the analysis. A more general case, when $\mathcal{D} = \mathcal{D}(\mathcal{E})$, was inspected by Knight and Peterson [18].

It is important to stress that, if the dependence $u(\mathcal{E})$ is not related to the specific shape of $j(E)$, shown in Figure 1, Equation (1) is nothing but a nonlinear diffusion equation of quite a general type describing a wide diversity of problems. Accordingly, the results discussed below may be applied to a much broader class of problems, provided these problems have traveling solitary-wave-type solutions.

For a steady-state traveling wave, the right-hand side of Equation (1) vanishes, and the equation transforms into an odinary differential equation. For the problem in question, a simple analysis reveals that its phase plane $(\mathcal{E}, \mathcal{E}_\xi)$ has three singular points situated at the $\mathcal{E}$ axis at $\mathcal{E} = \mathcal{E}_{1,2,3}$ corresponding to $E = E_{1,2,3}$ in Figure 1. Note that, by definition, $\mathcal{E}_2 \equiv 1$ since $\mathcal{E} = E/E_2$. In the phase plane, a single traveling domain is described by a homoclinic path beginning in the saddle $(\mathcal{E}_1, 0)$, making a loop around the unstable focus $(\mathcal{E}_2, 0)$ and ending up in the same saddle $(\mathcal{E}_1, 0)$, see Figure 3.

It is possible to show that such a solution of Equation (1) exists at $s = 1$ solely [18]. Since this is the only case I am interested in, $s$ below is always supposed to be equal to unity. Then, the homoclinic path may be found explicitly [18]; however, I do not need this expression for the subsequent inspection. Let us just designate the steady-state solution of Equation (1) as $\mathcal{E}_0(\xi)$. The goal of this paper is to analyze the stability of this solution against small time-dependent perturbations $\delta\mathcal{E}(\xi, \eta)$.

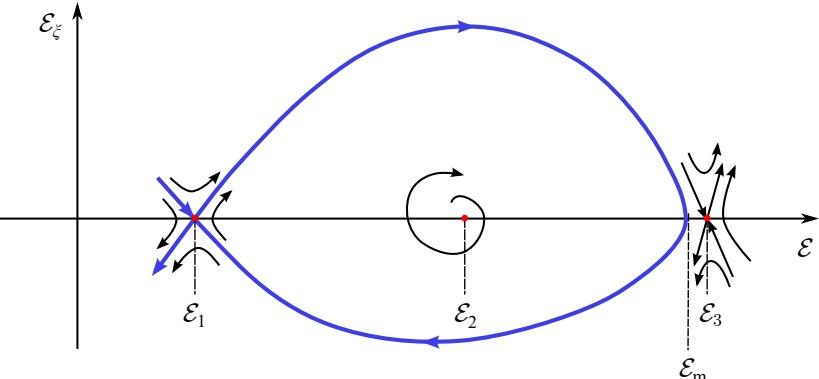

**Figure 3.** Phase plane $(\mathcal{E}, \mathcal{E}_\xi)$ (schematically). Three singular points are marked with red. The blue curve designates the homoclinic path corresponding to a single traveling domain. If the point $(\mathcal{E}_\mathrm{m}, 0)$ merges with $(\mathcal{E}_3, 0)$, the homoclinic path is split into two independent heteroclinic ones (the upper and lower parts of the homoclinic path, respectively). See text for details.

### 3. Stability Analysis

The stability analysis performed in Ref. [18] generalizes a brilliant approach by Zeldovich and Barenblatt for the inspection of the stability of a slow combustion front [24]. The main idea is as follows. Let us suppose that $\delta\mathcal{E}(\xi, \eta) = \mathcal{E}^{(1)}(\xi)\exp(-\lambda\eta)$, where $\lambda$ is an eigenvalue of the stability problem. If there is a negative $\lambda$ in the problem's spectrum, it means instability.

Substituting $\mathcal{E}(\xi, \eta) = \mathcal{E}^{(0)}(\xi) + \mathcal{E}^{(1)}(\xi)\exp(-\lambda\eta)$ in Equation (1) and linearizing the result in small $\mathcal{E}^{(1)}$, one arrives at the eigenvalue problem:

$$D\mathcal{E}^{(1)}_{\xi\xi} + \alpha[1 - u(\mathcal{E}^{(0)}(\xi))]\mathcal{E}^{(1)}_\xi - u_\mathcal{E}(\mathcal{E}^{(0)}(\xi))\mathcal{E}^{(1)} = -\lambda\mathcal{E}^{(1)}, \qquad (2)$$

supplemented with the boundary conditions $\mathcal{E}^{(1)} \to 0$ at $\xi \to \pm\infty$. Then, introducing a new function $\psi(\xi)$ connected with $\mathcal{E}^{(1)}(\xi)$ by the relation,

$$\psi(\xi) = \exp\left(\frac{\alpha}{2D}\int(1 - u(\mathcal{E}^{(0)}(\xi)))d\xi\right)\mathcal{E}^{(1)}(\xi) \equiv F(\xi)\mathcal{E}^{(1)}(\xi), \qquad (3)$$

one reduces Equation (2) to the standard Schrödinger equation:

$$\hat{H}\psi = \Lambda\psi, \qquad (4)$$

$$\hat{H} = -\frac{d^2}{d\xi^2} + V(\xi), \qquad (5)$$

$$V(\xi) = \left(\frac{\alpha(1 - u(\mathcal{E}^{(0)}(\xi)))}{2D}\right)^2 + \left(1 - \frac{\alpha\mathcal{E}^{(0)}_\xi(\xi)}{2}\right)\frac{u_\mathcal{E}(\mathcal{E}^{(0)}(\xi))}{D}, \qquad (6)$$

where $\Lambda \equiv \lambda/D$. Let us remark that there is a misprint in the expression for $V(\xi)$ in Ref. [19] corrected in Equation (6).

Note that, since the homoclinic path begins and ends up at the same singular point $(\mathcal{E}_1, 0)$ and $u(\mathcal{E}_1) = 1$, the considered steady-state traveling domain solution satisfies the condition $u(\mathcal{E}^{(0)}(\xi)) \to 1$ at $\xi \to \pm\infty$. Therefore, as it follows from Equation (3), $\psi(\xi)$ and $\mathcal{E}^{(1)}(\xi)$, both have the same asymptotic behavior at $\xi \to \pm\infty$. This is important since it means that none of the solutions of the Schrödinger equation generate "false" solutions of the initial stability problem, which may not satisfy the boundary conditions $\mathcal{E}^{(1)}(\xi) \to 0$ at $\xi \to \pm\infty$.

Now, the most essential part of the stability analysis begins. If $\mathcal{E}^{(0)}(\xi)$ is a solution of the steady-state version of Equation (2), then, owing to the translational invariance of the problem $\mathcal{E}^{(0)}(\xi + \xi_0)$, where $\xi_0$ is *any* constant, also is its solution, i.e., being substituted in the left-hand side of Equation (2), $\mathcal{E}^{(0)}(\xi + \xi_0)$, turns it to zero *identically*.

Let us consider the limit $\xi_0 \to 0$. In this case, $\mathcal{E}^{(0)}(\xi + \xi_0) \approx \mathcal{E}^{(0)}(\xi) + \mathcal{E}_\xi^{(0)}(\xi)\xi_0$, and $\mathcal{E}_\xi^{(0)}(\xi)\xi_0$ here may be regarded as an infinitesimal perturbation to $\mathcal{E}^{(0)}(\xi)$. The perturbation transforms the steady-state solution into another steady-state solution. This means that such a perturbation is *neutrally-stable* and should not evolve in time. In other words, it means that $\mathcal{E}_\xi^{(0)}(\xi)$ is an eigenfunction of the stability problem with zero eigenvalue.

Note that we obtain this result based on the translational invariance solely, without the employment of a specific form of the differential operator in Equation (2). These neutrally-stable modes generated by a transformation of a continuous group of symmetry are called *Goldstone* modes. Since 2008, when the implementation of such a mode in strong-interaction physics (do you remember my remark about interconnections of different fields in physics?) resulted in the Nobel Prize being awarded to Prof. Yoichiro Nambu, they have also been called *Nambu–Goldstone* modes.

It is interesting to note that, twenty-five years after the publication of Ref. [1], I returned to the inspection of the role of Goldstone modes in stability problems. This study resulted in the discovery of a new type of chaos at the onset analogous to the second-order phase transitions in statistical physics, where the mean amplitudes of the turbulent modes played the role of the order parameter [25–27].

However, I have departed from the stability analysis of the Gunn domain. It is high time to be back. Actually, not so much remains to be done. Collecting together all mentioned above, one can conclude that

$$\psi(\xi) = F(\xi)\mathcal{E}_\xi^{(0)}(\xi) \tag{7}$$

is the eigenfunction of the Schrödinger equation, Equations (4)–(6) with zero eigenvalue.

Recall now the oscillation theorem [20]. The theorem states that, in a one-dimensional Schrödinger equation, the $n$th wave function of a discrete spectrum should vanish $n$ times. Then, it is not a complicated task to show that the integral in the exponent in Equation (3) always remains finite, i.e., $F(\xi)$ never vanishes. Thus, all zeros of $\psi(\xi)$, if any, coincide with those of $\mathcal{E}_\xi^{(0)}(\xi)$. Finally, since for the traveling domain the profile $\mathcal{E}^{(0)}(\xi)$ has a single maximum (the homoclinic path in the phase plane $(\mathcal{E}, \mathcal{E}_\xi)$ crosses the $\mathcal{E}$-axis at $\mathcal{E} = \mathcal{E}_\mathrm{m}$, situated in between $\mathcal{E}_2$ and $\mathcal{E}_3$, see Figure 3), the product $F(\xi)\mathcal{E}_\xi^{(0)}(\xi)$ has a single zero at the value of $\xi$ corresponding to the maximal field achieved in the domain, $\mathcal{E}_\mathrm{m}$. It means that the wave function (3) is the one for the first excited state. The "energy" of the ground state should be lower than those for excited states. Since the first excited state has zero "energy" this gives rise to the conclusion that the ground state has negative "energy", i.e., the spectrum of Equations (4)–(6) has a single negative eigenvalue. In other words, the solution $\mathcal{E}^{(0)}(\xi)$ is *unstable* with the instability increment equal to the modulus of $\lambda$, corresponding to the ground state of the Schrödinger equation.

Though the problem in question has several parameters, the actual control parameter, which relatively easily may be varied in an experiment, is the current in the circuit, $j_\mathrm{ext}$. Varying $j_\mathrm{ext}$, one can change the values of $E_{1,2,3}$ and hence the shape of the traveling domain.

At a certain value of $j_\mathrm{ext} = j_0$, the maximal field in the domain, $\mathcal{E}_\mathrm{m}$ merges with $\mathcal{E}_3$, and the homoclinic path in the plane $(\mathcal{E}, \mathcal{E}_\xi)$ is split into two independent heteroclinic paths connecting the singular points $(\mathcal{E}_1, 0)$ and $(\mathcal{E}_3, 0)$. One of these paths lies entirely in the upper semi-plane. For this solution, $\mathcal{E}_\xi^{(0)}$ is always positive at any finite $\xi$. The other lies entirely in the lower semi-plane and, for it, $\mathcal{E}_\xi^{(0)} < 0$ at any finite $\xi$, see Figure 3. Each of these solutions corresponds to a traveling charge layer transferring the sample from one steady-state to another steady-state.

It is important that, since, for the layers, $\mathcal{E}_\xi^{(0)}$ does not vanish at any finite $\xi$, the corresponding wave function given by Equation (7) is the one of the *ground* state of the Schrödinger equation. This means that, in contrast to the traveling domain, the traveling layers are *stable* [18,19].

## 4. Spectrum of Schrödinger Equation

Thus, to get the value of the instability increment and, hence, the characteristic time for the traveling domain decomposition, one has to obtain the energy level for the ground state in the Schrödinger equation with the complicated potential given by Equation (6). As it has been mentioned above, the main idea employed in Ref. [1] to fulfill this task is to approximate the actual smooth potential by a superposition of square potential wells. The parameters of the wells are selected so that the approximated potential keeps all main features of the initial smooth profile. One of the approximation parameters remains free. It is fixed by the condition that, for the first excited state, $\Lambda = 0$.

This procedure reduces the solution of the complicated initial problem to finding the roots of a set of transcendental equations. In the worst case, the latter may be readily done numerically. To illustrate this rather general approach, a simple case of a broad traveling domain is discussed below.

The domain becomes broad when $j_{\text{ext}}$ approaches $j_0$. In the phase plane, it corresponds to the shift of the regular point of the path $(\mathcal{E}_{\text{m}}, 0)$ toward the saddle $(\mathcal{E}_3, 0)$. In this case, the domain approximately may be presented as a nonlinear superposition of two layers with opposite charges, and the potential (6) has a shape of two wells separated by a barrier.

Each well is associated with the corresponding layer. The width of the barrier equals the distance between the layers and is large in the case under consideration. Then, the tunneling through the barrier is exponentially weak. If the tunneling were suppressed entirely, each well would have corresponded to the potential generated by a single layer and, in accordance with the mentioned above, had the ground state with $\Lambda = 0$. Thus, it is clear that the level with $\Lambda < 0$ for the domain occurs due to the finite tunneling resulting in the splitting of the ground states in the two wells with the same value of $\Lambda = 0$. Therefore, instead of the employment of a rather cumbersome general procedure of the solution of an entire problem with the finite tunneling, let us, first, neglect the tunneling and consider the solutions of the Schrödinger equation in each square well separately. Then, the finite tunneling is taken into account with the help of perturbation theory.

The values of the parameters of the square wells approximating the smooth profile of the potential may be obtained by inspection of Equation (6). For example, bearing in mind that at $\xi \to \pm\infty$, the solution $\mathcal{E}^{(0)}(\xi)$ describing the domain satisfies the conditions $\mathcal{E}^{(0)} \to \mathcal{E}_1$; $\mathcal{E}_\xi^{(0)} \to 0$ and that, by definition, $u(\mathcal{E}_{1,2,3}) = 1$, one immediately obtains that both outer walls of the wells have a height equal to $u_{\mathcal{E}}(\mathcal{E}_1)/\mathcal{D}$. Similarly, the barrier height is $u_{\mathcal{E}}(\mathcal{E}_{\text{m}})/\mathcal{D} \approx u_{\mathcal{E}}(\mathcal{E}_3)/\mathcal{D}$. The widths of the wells and the barrier are estimated based on the exact solution describing the domain path in the phase plane obtained in Ref. [18]. The last remaining parameters are the depths of the wells. They are fixed by the conditions that the ground state in each well has $\Lambda = 0$.

I do not present here these simple but cumbersome calculations. Just note that the problem is rather robust against errors in the approximation of $V(\xi)$. The robustness is related to the smallness of the split of the ground levels in the wells due to the tunneling and the fact that, at the employed approach, the important condition $\Lambda = 0$ for the ground state in each separate well holds automatically.

Let us suppose that the wave functions, $|1, 2\rangle$, of the ground state for each well are known and that these wave functions satisfy the equations $\hat{H}_{1,2}|1, 2\rangle = 0$. Here, $\hat{H}_{1,2}$ designates the Hamiltonians, whose potentials, $U_{1,2}$ are the corresponding single-well potentials. Let us look for the wave function of the complete problem with the two-well potential in a form of a linear superposition of $|1, 2\rangle$:

$$|\psi\rangle = c_1|1\rangle + c_2|2\rangle, \tag{8}$$

where $c_{1,2}$ are constants, which should be defined in the course of calculations. Then, since $|\psi\rangle$ is an eigenfunction of the complete Hamiltonian $\hat{H}$,

$$c_1\hat{H}|1\rangle + c_2\hat{H}|2\rangle = \Lambda(c_1|1\rangle + c_2|2\rangle). \tag{9}$$

Making scalar products with $\langle 1, 2|$ and taking into account the normalization conditions $\langle 1|1\rangle = \langle 2|2\rangle = 1$, one arrives from Equation (9) to the following equations for $c_{1,2}$:

$$(H_{11} - \Lambda)c_1 + (H_{12} - \Lambda\langle 1|2\rangle)c_2 = 0, \tag{10}$$
$$(H_{21} - \Lambda\langle 2|1\rangle)c_1 + (H_{22} - \Lambda)c_2 = 0, \tag{11}$$

where $H_{11}$, $H_{12}$, $H_{21}$ and $H_{22}$ stand for the corresponding matrix elements. Note that the wave functions $|1\rangle$ and $|2\rangle$ are *not orthogonal* since they are the eigenfunctions of the *different* Hamiltonians, namely, $\hat{H}_1$ and $\hat{H}_2 \neq \hat{H}_1$.

The solvability condition requires vanishing of the determinant of Equations (10) and (11). This results in a quadratic equation for $\Lambda$. The difference, $\Delta = |\Lambda_1 - \Lambda_2|$, between the two roots of this equation approximately equals the desired instability increment.

This result has completed the instability analysis. However, the exact expression for $\Delta$ is rather cumbersome. Therefore, it is worth simplifying this result, employing the smallness of certain parameters in Equations (10) and (11). To this end, I have to estimate the matrix elements and the overlap integrals $\langle 1, 2|2, 1\rangle$.

To calculate the matrix elements, it is convenient to single out from the full double-well square potential, $V_{\mathrm{DWS}}(\xi)$, the part corresponding to a single well, i.e., to suppose that $V_{\mathrm{DWS}}(\xi) = U_{1,2} - V_{1,2}$, see Figure 4, where for the first well (left side of Figure 4), $V_1 = 0$ at $\xi < \xi_3$, while, at $\xi > \xi_3$, the sum of $V_{\mathrm{DWS}}$ and $V_1$ equals the height of the barrier. Similarly, for the second well (right), the sum $V_{\mathrm{DWS}} + V_2$ equals the height of the barrier at $\xi < \xi_2$, while, at $\xi > \xi_2$, the potential $V_2 = 0$. Thus, $U_{1,2}$ are the single-well potentials, and the corresponding Hamiltonians acting on their wave functions produce zero. Then, the leading terms in the matrix elements are estimated as follows:

$$H_{11} = -\langle 1|V_1|1\rangle \sim H_{22} = -\langle 1|V_2|1\rangle \sim \exp[-2d_{\mathrm{b}}\sqrt{u_{\mathcal{E}}(\mathcal{E}_3)/\mathcal{D}}], \tag{12}$$

where $d_{\mathrm{b}} = \xi_3 - \xi_2$ is the barrier width. The same estimate is true for $H_{22}$ (for the sake of simplicity, the widths of the corresponding wells, $d_{1,2}\sqrt{u_{\mathcal{E}}(\mathcal{E}_3)/\mathcal{D}}$, are supposed to be not small). In the same manner, one obtains

$$H_{12} \sim H_{21} \sim \langle 2|1\rangle \sim \langle 1|2\rangle \sim \exp[-d_b\sqrt{u_{\mathcal{E}}(\mathcal{E}_3)/\mathcal{D}}], \tag{13}$$

Then, in the leading approximation,

$$\Delta \approx H_{12} + H_{21} \sim \exp[-d_b\sqrt{u_{\mathcal{E}}(\mathcal{E}_3)/\mathcal{D}}]. \tag{14}$$

Thus, as it could be expected, the value of the instability increment for the broad domain is exponentially small indeed.

Finally, note that, for each single square well, the normalized wave functions $|1, 2\rangle$ may be readily obtained in the explicit form, including the normalization constant. Then, it is just a matter of more or less routine calculations to improve the accuracy of Equation (14), taking into account the prefactors and dropped higher-order exponentially small terms.

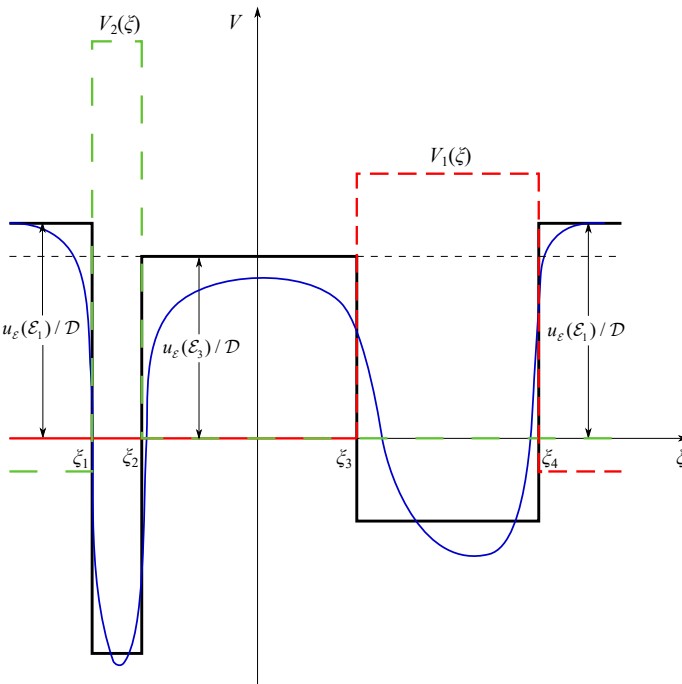

**Figure 4.** Schematically: The actual double-well potential $V(\xi)$ (smooth blue line). The approximation of $V(\xi)$ by the double-well square potential, $V_{\text{DWS}}(\xi)$, is shown in black; $\xi_{1,2,3,4}$ designate the coordinates of the walls of the wells. $U_{1,2}(\xi) = V_{\text{DWS}}(\xi) + V_{1,2}(\xi)$ are the potentials of the single-well approximation, when tunneling is neglected.

## 5. Conclusions

Summarizing and generalizing the discussed above, one arrives at the following conclusions:

- The analysis of the linear stability of traveling wave solutions in a wide class of nonlinear diffusion problems is reduced to inspection of a bottom part of the spectrum of the associated Schrödinger equation, whose potential is generated by the profile of the analyzed solution.
- The translational invariance transformation generates in the stability spectrum a neutrally-stable (Goldstone) mode.
- The qualitative answer to the question about the stability of the solution is readily obtained based on the oscillation theorem—if the Goldstone mode does not have any nodes, the solution is stable. Otherwise, it is unstable.
- To quantitatively characterize the instability (if any), the "energy" level of the ground state of the Schrödinger equation should be obtained.
- A powerful tool to make the problem of a bottom part of the Schrödinger equation spectrum tractable is to approximate the potential by square wells.

These conclusions are rather general. They are valid far beyond the frameworks of the Gunn effect and, hopefully, may help to analyze the stability of traveling waves in a broad class of nonlinear diffusion problems.

The developed approach to find an approximate solution and spectrum of the Schrödinger equation with a complicated potential valid for ground and low-excited states may be regarded as a complement to the known Wentzel-Kramers-Brillouin (WKB) method, which is good in the opposite case of high-excited states. A common disadvantage of both approaches is the approximation error, which is difficult to improve and even to control.

**Funding:** This work is financed by the Moscow Engineering Physics Institute Academic Excellence Project (in agreement with the Ministry of Education and Science of the Russian Federation of 27 August 2013, Project No. 02.a03.21.0005).

**Acknowledgments:** I am very grateful to the entire Editorial staff of *Physics* for this Special Issue dedicated to my humble person. My special thanks to my friends and colleagues Boris Malomed, Andrey Miroshnichenko, and Fernando Moreno, who volunteered to be Guest Editors of the Special Issue and completed this difficult task in an excellent manner. Thank you very much indeed!

**Conflicts of Interest:** The author declares no conflict of interest.

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
