# Peer review of "Instability of Traveling Pulses in Nonlinear Diffusion-Type Problems and Method to Obtain Bottom-Part Spectrum of Schrödinger Equation with Complicated Potential"

_2624-8174, doi:10.3390/physics3030043_

Round 1

Reviewer 1 Report

The paper presents an elegant solution to a stability problem put into historical context. The paper is instructive and educative to the readers.

Author Response

Many thanks for the high rating of the paper.

Reviewer 2 Report

Please see my report in the attached pdf file. I make suggestions to render the paper more readable and understandable as well as adding two relevant references.

Author Response

Many thanks for the careful reading of the MS and very helpful constructive comments. For my detailed reply to every issue raised in your report see the uploaded file Cover_letter_F.pdf

Round 2

Reviewer 2 Report

The author corrected the paper taking into account my suggestions and minor corrections. Thus, the paper is now fine.